

# Dual applications of Chebyshev polynomials method: Efficiently finding thousands of central eigenvalues for many-spin systems

**Haoyu Guan and Wenxian Zhang**⋆

Key Laboratory of Artificial Micro- and Nano-structures of Ministry of Education,
and School of Physics and Technology, Wuhan University, Wuhan, Hubei 430072, China

⋆ wxzhang@whu.edu.cn

## Abstract

Computation of a large group of interior eigenvalues at the middle spectrum is an important problem for quantum many-body systems, where the level statistics provides characteristic signatures of quantum chaos. We propose an exact numerical method, dual applications of Chebyshev polynomials (DACP), to simultaneously find thousands of central eigenvalues, where the level space decreases exponentially with the system size. To disentangle the near-degenerate problem, we employ twice the Chebyshev polynomials, to construct an exponential semicircle filter as a preconditioning step and to generate a large set of proper basis states in the desired subspace. Numerical calculations on Ising spin chain and spin glass shards confirm the correctness and efficiency of DACP. As numerical results demonstrate, DACP is 30 times faster than the state-of-the-art shift-invert method for the Ising spin chain while 8 times faster for the spin glass shards. In contrast to the shift-invert method, the computation time of DACP is only weakly influenced by the required number of eigenvalues, which renders it a powerful tool for large scale eigenvalues computations. Moreover, the consumed memory also remains a small constant (5.6 GB) for spin-1/2 systems consisting of up to 20 spins, making it desirable for parallel computing.

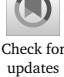

# 1  Introduction

Energy level statistics provides an essential characterization of quantum chaos [1, 2]. Integrable systems often imply level clustering and a Poisson distribution of energy level spacings [2], while chaotic systems exhibit level repulsion and a Wigner-Dyson distribution [3]. Other useful statistical tools like the $\delta_3$ statistic [4] and the power spectrum of the $\delta_n$ statistic also depend on the level distribution [5]. Numerical eigenvalues are important to exactly characterize these level statistics, because analytical results are not available in general. In addition, individual eigenstates at the middle of the spectrum, which correspond to the "infinite temperature" limit, are of great importance in studying the many-body localization (MBL) [6–8]. Numerical simulations are a major tool in understanding quantitatively many aspects of the MBL problem. As the many-body problems of interest often involve a huge Hilbert space whose dimension grows exponentially with the system size, it is rather challenging to fully diagonalize the Hamiltonian or to solve the time-independent Schrödinger equation. Seeking to resolve the eigenvalue problem in a small part of the spectrum is thus an unavoidable and desirable substitution.

To obtain the interior eigenstates, a hybrid strategy of matrix spectroscopy is often invoked [9–11]. It aims at computing eigenstates in selected regions of the spectrum and combines the ground state solvers with a spectral filter, where the filter is designed to transform the selected interior regions to the edges of the spectrum. Among these filters, the Green function $(\mathcal{H}-\lambda I)^{-1}$ is an excellent one. After applying the Green function filter, the cluster of eigenvalues near the energy $\lambda$ is mapped to very large positive and negative values. The level spacings near $\lambda$ are amplified, which improves the convergence. The Lanczos method [12] was combined with such a filter [9,10], and the Chebyshev polynomial expansion of the Green function is implemented [13]. In particular, the shift-invert method [14] essentially utilizes this spectral transformation and is widely used in quantum many-spin systems [8,15–18]. Moreover, it was considered to be the most efficient one for the MBL problems [8]. However, for large systems this method suffers rapid increases in computation time and memory consumption, due to the factorization of $(\mathcal{H}-\lambda I)^{-1}$. Other spectral filters have also been proposed, including the Dirac delta function filter $\delta(\mathcal{H}-\lambda I)$ [19–23] and the rectangular window function filter [24].

Other more efficient methods were proposed to compute large numbers of eigenpairs located at the interior spectrum. In particular, the eigenvalues slicing library (EVSL) adopts the Chebyshev expansion of the Dirac delta function during the polynomial filtering process, combined with the restarted Lanczos algorithm [21]. Similarly, FILTLAN uses a combination of

(non-restarted) Lanczos and polynomial filtering with Krylov projection methods to calculate both the interior and extreme eigenvalues [20]. The FEAST algorithm exploits the well-known Cauchy integral formula to express the eigenprojector, leading to a rational filter to which subspace iteration is then applied [25]. Although these methods are capable of finding thousands or tens of thousands eigenpairs using a *divide and conquer* strategy, they can calculate only several hundreds or a thousand eigenpairs in a single energy interval. Note that all the methods mentioned above are iterative. The computation time is approximately proportional to the required number of eigenvalues, i.e., more eigenvalues requires more filtrations and re-orthogonalizations [8, 20, 22].

In this paper, we propose an exact numerical method, DACP, to calculate thousands of eigenvalues at the middle of the energy band, which is enough to reveal the level statistics [26, 27]. For spin systems, such a middle region usually indicates a peak of the density of states where the energy levels are nearly degenerate. It is extremely challenging to distinguish these near-degenerate eigenvalues without factorizing $\mathcal{H}^{-1}$. In the DACP method, we construct an exponential of semicircle (exp-semicircle) filter to quickly damp the unwanted part of the spectrum by employing the Chebyshev polynomial for the first time. The second application of the Chebyshev polynomial is to fast search a set of states to span the specific subspace, which consists of all the desired eigenstates. Combining these two steps, the DACP method essentially transforms the original high-dimension eigenvalue problem to a low-dimension one. Instead of many iterative filtrations, the DACP method directly produces results with only a single filtration.

For practical problems in many-spin systems, the DACP method is very efficient, due to its full exploration of several excellent properties of the Chebyshev polynomial, while other methods utilize only a part of them. For a large class of many-spin systems, the DACP method exhibits a significant increase in computation speed, up to a factor of 30, in comparison with the shift-invert method. The memory saving is more drastic, up to a factor of 100. Moreover, the DACP method distinguishes itself from those iterative filtering ones, as its convergence time varies slightly when the required number of eigenvalues changes in a large region.

The paper is organized as follows. The detailed formalism of the DACP method, including the exp-semicircle filtration, Chebyshev evolution, and subspace diagonalization, is described in Section 2. Each of the processes relies on a particular property of the Chebyshev polynomial, distinguishing the DACP from other filtering methods . In Section 3 we calculate the interior eigenvalues with the DACP in two many-spin systems, the Ising model and the spin glass shards, and present the numerical results. We compare the DACP with other approaches in Section 4 and conclude in Section 5.

## 2 Dual applications of Chebyshev polynomials method

To access the central eigenvalues of large spin systems, naturally we are restricted by the matrix-free mode, i.e., the matrix of the Hamiltonian must not be explicitly expressed/stored. Instead, we treat the Hamiltonian $\mathcal{H}$ as an operator whose input and output are states/vectors. Therefore, we shall only operate with the set of quantum states:

$$\left\{ |\psi\rangle, \mathcal{H}|\psi\rangle, \mathcal{H}^2|\psi\rangle, \cdots, \mathcal{H}^k|\psi\rangle \right\}, \tag{1}$$

where $k$ is a positive integer, in the original Hilbert space of dimension up to $5 \times 10^5$. Note that each "matrix-vector product" $\mathcal{H}|\psi\rangle$, being a basic operation, consumes considerable time while only a small amount of memory. Moreover, as illustrated in Subsection 2.3, even the states in Eq. (1) are not necessarily required to be stored. Therefore, the requirement of memory in DACP is pretty tiny. In general, we hope to simultaneously find a large scale of

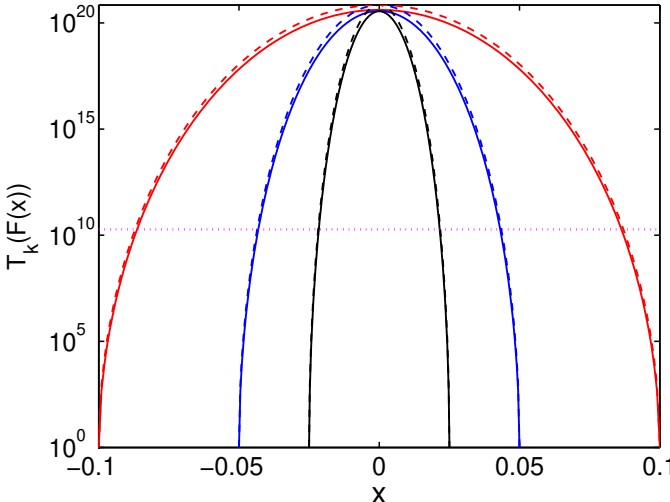

Figure 1: The exp-semicircle filters for $a = 0.1$ (red lines), $0.05$ (blue lines), and $0.025$ (black lines), with $ka = 24$. We have set $E_{\max} = 1$ and $E_{\min} = -1$. The solid lines are for the Chebyshev polynomials $T_k(F(x))$, where $F(x) = \left[2x^2 - (1+a^2)\right]/(1-a^2)$, while the dashed lines are for the approximations $y = \exp(2k\sqrt{a^2-x^2}) \simeq T_k(F(x))$. The horizontal pink dotted line denotes the half maximum of the logarithm of filters. Note that $|T_k(F(x))| \leq 1$ when $x \notin [-a, a]$, thus the filter regions are restricted in $[-a, a]$.

central eigenvalues. In this manuscript, we set our goal to find $5{,}000$ eigenvalues accurately at the middle of the spectrum for the many-spin systems as an illustration.

The idea of DACP is fairly straightforward. We first transform a randomly initialized state into a wave packet in the subspace spanned essentially by at least $5{,}000$ central eigenstates. This transformation is realized by an exp-semicircle filter implemented by the Chebyshev polynomial. Here the Chebyshev polynomial is used to realize an exponential decay. With this particular state in hand, we then generate a large amount of linearly independent states, as large as possible, to approximately span the subspace consisting of the required eigenstates. The Chebyshev polynomial is used again to oscillate (complex exponential) the states, producing approximately linearly independent states. Once the generating set for the desired eigenstates is known, one may explicitly calculate $H$, the reduced representation of $\mathcal{H}$ in this subspace. The remaining operations are restricted in the subspace of dimension around $10^4$. Finally, direct diagonalization of $H$, which is of size $10^4 \times 10^4$, gives the desired eigenvalues. The detailed procedures and discussions are given in the subsections below.

## 2.1 Exp-semicircle filter

We utilize the exponential growth of the Chebyshev polynomials outside the interval $[-1, 1]$ to efficiently construct an exp-semicircle filter, as shown in Figure 1. The filter drastically amplifies the components of a desired range of eigenstates for any randomly initialized states, resulting a new state that sharply localized at the middle of the spectrum. We note that the Chebyshev filter explores the same property as well, except that it amplifies the lower end of the spectrum [28, 29]. A similar idea was applied in the quantum algorithm for finding ground states [30].

For the Hamiltonian $\mathcal{H}$ with energy bounded in $[E_{\min}, E_{\max}]$, where $E_{\min}$ is the minimum ($E_{\min} < 0$) and $E_{\max}$ the maximum ($E_{\max} > 0$), the exp-semicircle filter is designed to amplify the components of the eigenstates corresponding to eigenvalues in the interval $[-a, a]$ and to

simultaneously dampen those in the interval $[E_{\min}, -a]$ and $[a, E_{\max}]$, where $a$ is a real positive parameter. Focusing on the spin systems, we have assumed $E_{\min} < -a$ and $a < E_{\max}$. After the filtration, a new state mainly consisting of the eigenstates with eigenvalues belonging to the interval $[-a, a]$ is generated. For simplicity, we denote the subspace spanned by the eigenstates contained in $[-a, a]$ as $\mathbb{L}$. Consequently, the probabilities of eigenstates outside $\mathbb{L}$ are almost negligible, as can be seen in Figures 1 and 2.

We now introduce the specific implementation details. Note that we want to amplify the middle of the spectrum, but the exponential growth of the Chebyshev polynomial exists only near both ends. To circumvent this difficulty, we first square the Hamiltonian, obtaining $\mathcal{H}^2$ with a spectrum ranges $[0, E_{\max}^2]$ (suppose $E_{\min} = -E_{\max}$ for simplicity). The middle spectrum $[-a, a]$ of $\mathcal{H}$ is transferred to $[0, a^2]$ for $\mathcal{H}^2$, which lies exactly at the lower end. Next is to map the dampening part $[a^2, E_{\max}^2]$ into $[-1, 1]$ by shift and normalization of $\mathcal{H}^2$. We thus define an operator

$$\mathcal{F} = \frac{\mathcal{H}^2 - E_c}{E_0}, \tag{2}$$

where $E_c = (E_{\max}^2 + a^2)/2$ and $E_0 = (E_{\max}^2 - a^2)/2$. One may easily affirm this map's correctness by replacing $\mathcal{H}^2$ with either $a^2$ or $E_{\max}^2$, and correspondingly one has $F(x) = (x^2 - E_c)/E_0$. Note that $\mathcal{F}$ is simply a polynomial expression of $\mathcal{H}$, so is $T_k(\mathcal{F})$.

We then explore the effect of the filtration using $T_k(\mathcal{F})$. As the eigenvalues inside $[0, a^2]$ of $\mathcal{H}^2$ are mapped into $[-1 - 2a^2/(E_{\max}^2 - a^2), -1]$ of $\mathcal{F}$, we obtain $T_k(\mathcal{F}) = (-1)^k \cosh(k\Theta)$ for the lower end of the spectrum, where

$$\Theta = \cosh^{-1}\left(1 + \frac{2(a^2 - \mathcal{H}^2)}{E_{\max}^2 - a^2}\right). \tag{3}$$

Let $|\psi\rangle = \sum_i c_i |\phi_i\rangle + \sum_j d_j |\chi_j\rangle$ be a random initial state, with $c_i$ and $d_j$ the random coefficients, $|\phi_i\rangle$ the eigenstates inside $\mathbb{L}$, $|\chi_j\rangle$ the eigenstates outside $\mathbb{L}$. The filtration by $T_k(\mathcal{F})$ is

$$
\begin{aligned}
|\psi(k)\rangle &= T_k(\mathcal{F})|\psi\rangle \\
&= \sum_i \left(e^{k\theta_i^{\text{in}}} + e^{-k\theta_i^{\text{in}}}\right) \frac{c_i}{2} |\phi_i\rangle + \sum_j \left(e^{ik\theta_j^{\text{out}}} + e^{-ik\theta_j^{\text{out}}}\right) \frac{d_j}{2} |\chi_j\rangle \\
&\simeq \frac{1}{2} \sum_i e^{k\theta_i^{\text{in}}} c_i |\phi_i\rangle,
\end{aligned}
\tag{4}
$$

where $\theta_i^{\text{in}} = \cosh^{-1}(1 + 2(a^2 - E_i^2)/(E_{\max}^2 - a^2))$, $\theta_j^{\text{out}} = \cos^{-1}(2(E_j^2 - a^2)/(E_{\max}^2 - a^2) - 1)$, $E_i$ and $E_j$ are eigenvalues corresponding to $|\phi_i\rangle$ and $|\chi_j\rangle$, respectively. In writing Eq. (4) we have ignored $(-1)^k$, as it does not affect the absolute value of coefficients and is a global phase at the end line. When $a$ is tiny, i.e., $a^2 \ll E_{\max}^2$, one may further deduce $\theta_i^{\text{in}} \simeq 2\sqrt{a^2 - E_i^2}/E_{\max}$ via Taylor's expansion of $\cosh^{-1}(1 + \varepsilon)$, where $\varepsilon$ is a small positive number. We thus obtain the exp-semicircle filter

$$T_k(\mathcal{F}) \simeq e^{\frac{2k}{E_{\max}}\sqrt{a^2 - \mathcal{H}^2}} \tag{5}$$

that peaks sharply at $E_i = 0$ with a large $k$, for eigenstates satisfying $-a \leq E_i \leq a$. In Figure 1 the shape of Eq. (5) is presented in dashed lines, which agrees well with the exact results.

With the initial conditions $T_0(\mathcal{F}) = 1$ and $T_1(\mathcal{F}) = \mathcal{F}$, the $k$th order Chebyshev polynomial can be efficiently determined using the recurrence relation Eq. (A.3). In this paper, we set the cut-off order $K = 12E_{\max}/a$. Such a filter exponentially (the fastest rate among all polynomials) amplifies the components of eigenstates inside $\mathbb{L}$ [31]. After the normalization, it equivalently dampens those outside $\mathbb{L}$, generating the target state:

$$|\psi_E\rangle \simeq \sum_i c_i' |\phi_i\rangle, \tag{6}$$

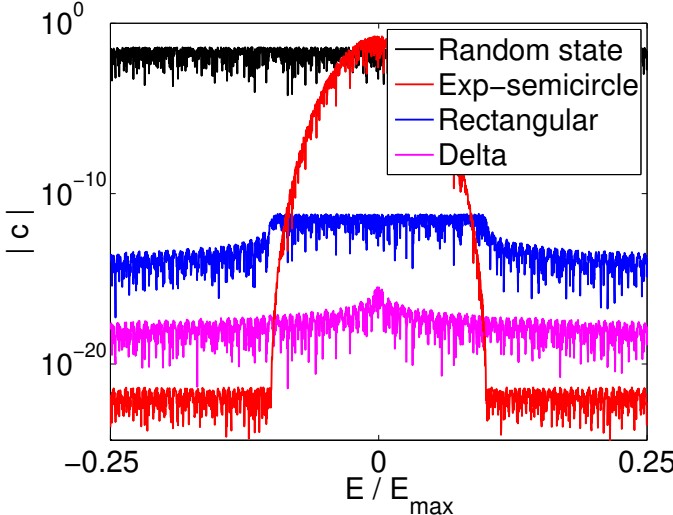

Figure 2: Comparison of the effect of three different filters in the interval $[-0.25, 0.25]$. We present the initial random normalized state $|\psi\rangle = \sum_i c_i |\phi_i\rangle$ (black curve) and the final normalized states after the filtration by the exp-semicircle filter with $K = 240$ (red curve) and the Chebyshev expansion of the rectangular function $\Theta(x+0.1)\Theta(-x+0.1)$ with order $K = 480$ (blue curve) and the Dirac delta function $\delta(0)$ with $K = 480$ (pink curve). The parameter $a = 0.1$ for the exp-semicircle filter. The $x$-axis corresponds to the "reduced energy" $E/E_{\max}$, while the $y$-axis to the absolute value of the probability amplitudes $c_i$. The rectangular function is shifted by $10^{-10}$ while the delta function is shifted by $10^{-15}$. All three filters share the same number of matrix-vector products. The advantage of the exp-semicircle filter is obvious, considering the amplification of the target amplitudes.

where $c_i' = \beta e^{K\theta_i^{\text{in}}} c_i$ and $\beta$ is the normalization constant. Obviously, the state $|\psi_E\rangle$ localizes (in the energy representation) at the middle of the spectrum. We input $|\psi_E\rangle$ as the initial state for Subsection 2.2, Chebyshev evolution.

To further illustrate the advantage of the exp-semicircle filter, we compare it with the Chebyshev expansion of two other functions. The first one is the Dirac delta function $\delta(0)$, which was employed to efficiently find interior eigenvalues in Refs. [21–23, 32]. The second is the rectangular function, which was also used for interior eigenvalue computations [24]. Setting a randomly initialized state $|\psi\rangle = \sum_i c_i |\phi_i\rangle$, we plot the generated states after the filtration by the three filters in Figure 2. All three filters are realized with the same number of matrix-vector products. The target interval of the delta filter is quite small. However, for our purpose, it is the relative magnitude between the target interval ($[-0.1, 0.1]$) and the others (outside the target interval) that matters. Thus, the efficiency of the exp-semicircle filter is confirmed apparently.

## 2.2 Chebyshev evolution

After obtaining a state confined in the small subspace $\mathbb{L}$, we then make use of the oscillation property of the Chebyshev polynomial to efficiently generate a set of *distinct* states, as many as possible, serving as a complete basis to span $\mathbb{L}$. To achieve this goal, it is necessary to limit $E_i \in [-1, 1]$ (corresponding to $x$ in $T_k(x)$), within which the Chebyshev polynomial behaves as a cosine-like function. This region contrasts with the requirement of the exp-semicircle filter, thus one needs a different transformation of $\mathcal{H}$. Below we describe the specific details for the second application of the Chebyshev polynomial.

The original Hamiltonian $\mathcal{H}$ needs to be shifted by $E_c'$ and to be rescaled by $E_0'$, where $E_c' = \frac{1}{2}(E_{\min} + E_{\max})$ and $E_0' = \frac{1}{2}(-E_{\min} + E_{\max})$. In a similar way to Eq. (2), we define an operator $\mathcal{G} = (\mathcal{H} - E_c')/E_0'$, which is definitely bounded by $-1$ and $1$. Assuming $E_{\min} = -E_{\max}$ again, we obtain

$$\mathcal{G} = \frac{\mathcal{H}}{E_{\max}}. \tag{7}$$

In this manner, the parameter $a$ is rescaled as $a_r = a/E_{\max}$.

Let us explore the Chebyshev evolution, which is governed by the operator $T_k(\mathcal{G})$ as $k$ plays the role of time. We input the state Eq. (6) generated by the filtration as an initial state for the Chebyshev evolution. Since $\|\mathcal{G}\| \leq 1$, from Eq. (A.2) we have $T_k(\mathcal{G}) = \cos(k\Omega)$, where $\Omega = \arccos(\mathcal{G})$. In this sense, the Chebyshev evolution becomes

$$\begin{aligned}
|\psi_E(k)\rangle &= T_k(\mathcal{G})|\psi_E\rangle \\
&= \frac{1}{2}\sum_j \left(e^{ik\omega_j} + e^{-ik\omega_j}\right) c_j' |\phi_j\rangle,
\end{aligned} \tag{8}$$

where $\omega_j = \arccos(E_j/E_{\max})$. Note that both $\omega_j$ and $k$ are unitless. Certainly, this is not a physical evolution, and $T_k(\mathcal{G})$ is not even a unitary operator. Actually, this evolution essentially represents a superposition of both forward and backward propagation. Each time the polynomial order $k$ increases by 1, the evolution "time" is added by 1 as well.

With the aid of the Chebyshev evolution, we are able to construct a complete basis that spans the subspace $\mathbb{L}$. In detail, we collect a set of states as follows

$$\left\{\hat{I}, \sin(\hat{X}), \cdots, \sin(n\hat{X}), \cos(\hat{X}), \cdots, \cos(n\hat{X})\right\} |\psi_E\rangle, \tag{9}$$

where $\hat{X} = \pi\mathcal{G}/a_r$ and $n$ is an integer determined by the relation $2n + 1 \geq d$, with $d$ the dimension of $\mathbb{L}$. In practice, one may need a relation $2n + 1 = 1.5d$ to ensure the completeness of Eq. (9). Here $k \simeq m\pi/a_r$ serves as the time, with $m = 1, \cdots, n$. Similar to Subsection 2.1, the $k$th order Chebyshev polynomial $T_k(\mathcal{G})$ is calculated. The cut-off order (evolution time) $K' = \lfloor n\pi/a_r \rfloor$. More details can be found in Appendix B.

The duality of the Chebyshev polynomials, as being approximately the trigonometry functions and being the polynomials, plays a vital role in the DACP method. In order to distinguish those clustered eigenvalues to the utmost, we need a set of basis functions whose slopes are as steep as possible, which amounts to the violent oscillations. In a certain sense, the Chebyshev evolution is an efficient (possibly the most efficient among all the polynomials) simulation of the quantum oscillations [33]. Replacing the operator $\mathcal{G}$ with a real variable $x$, the maximum slope around $E = 0$ for Eq. (9) is $\frac{n\pi}{a_r}$, in sharp contrast to $\frac{1}{a_r}$ given by the (most common) operator set

$$\left\{\hat{I}, \frac{\mathcal{G}}{a_r}, \frac{\mathcal{G}^2}{a_r^2}, \cdots, \frac{\mathcal{G}^n}{a_r^n}\right\}. \tag{10}$$

We find that a steep slope is helpful in distinguishing near-degenerate eigenvalues. Moreover, under the circumstance of the simulations stated in Appendix E, using the basis Eq. (10), instead of Eq. (9), during the Chebyshev evolution exhibits rather poor convergence as shown in Figure 8. These facts clearly illustrate the advantages of the second application of the Chebyshev polynomials.

## 2.3 Subspace diagonalization

Computing the basis $\{|\Psi_i\rangle : i = 1, \cdots, 2n + 1\}$ (Eq. (9)), by combining the exp-semicircle filter and the Chebyshev evolution, represents the most challenging aspect as well as the most time-consuming part of the DACP method. Once the appropriate basis is constructed, the remaining

task is straightforward, i.e., to compute the eigenpairs of the projected Hamiltonian $H$. This is equivalent to solving the generalized eigenvalue problem

$$HB = SB\Lambda. \tag{11}$$

Here, $H$ and $S$ denote the projected Hamiltonian in $\mathbb{L}$ and the overlap matrices, respectively,

$$H_{ij} = \langle \Psi_i | \mathcal{H} | \Psi_j \rangle, \quad S_{ij} = \langle \Psi_i | \Psi_j \rangle. \tag{12}$$

$\Lambda$ is a diagonal matrix with the eigenvalues in $[-a, a]$ and the matrix $B$ transforms the found basis Eq. (9) to the eigenstates $|\phi_j\rangle$ of $\mathcal{H}$,

$$|\phi_j\rangle = \sum_{i=1}^{2n+1} B_{ij} |\Psi_i\rangle. \tag{13}$$

All these matrices are of size $(2n + 1) \times (2n + 1)$. Because of its small size, $H$ can be readily diagonalized by the LAPACK library [34].

Importantly, due to the special property of the Chebyshev polynomial, the computation of matrices $H$ and $S$ can even be achieved without an explicit computation and storage of the states $|\Psi_i\rangle$. This feature gives rise to a further improvement for the DACP method, both in computation time and memory. Besides, for an overcomplete basis Eq. (9), the overlap matrix $S$ is generally singular. We thus employ the singular value decomposition (SVD). After the SVD, the eigenvalues of $S$ are discarded if their absolute values sit below the cutoff condition $\varepsilon = 10^{-12}$. The number of remained eigenvalues effectively counts the linearly independent states in Eq. (9). We present both the explicit expressions (denoted by $T_k(\mathcal{G})$ and $|\psi_E\rangle$ only) of matrices $H$ and $S$, and the solution of the generalized eigenvalue problem in Appendix C.

Eigenvalues obtained by the subspace diagonalization may not own the same accuracy when compared to true eigenvalues of the system, thus it is necessary to conduct an independent check to estimate the error bounds. To this end, if the eigenstates $|\phi_j\rangle$ were known, the residual norm (variance of the energy) $||r_j|| = \sqrt{\langle \mathcal{H}^2 \rangle - \langle \mathcal{H} \rangle^2}$, where $\langle \mathcal{H}^2 \rangle = \langle \phi_j | \mathcal{H}^2 | \phi_j \rangle$ and $\langle \mathcal{H} \rangle = \langle \phi_j | \mathcal{H} | \phi_j \rangle$, is widely used as the parameter measuring the accuracy of the results. It has been shown that $||r_j||$ gives an upper bound on the true error (absolute error) of the computed eigenvalue [35].

## 3 Numerical results

We apply the DACP method to the quantum spin-1/2 systems with two-body interactions. Such systems are good models for investigating a large class of important problems in quantum computing, solid state theory, and quantum statistics [36–39]. A large number of exact eigenvalues help us to obtain the statistical properties, to distinguish quantum chaos from integrability, and serve as a benchmark to evaluate other approximate methods as well.

Generally speaking, the DACP method can deal with spin systems consisting of couplings between any pair of $N$ spins. Each of the Pauli matrix $\sigma^\alpha$ or the two coupling Pauli matrices $\sigma^\alpha \otimes \sigma^\beta$, where $\alpha, \beta = x, y, z$, is properly represented by a specific function.

We specify the spin model for two physical systems. One is the disordered one-dimensional transverse field Ising model [40], where the Hamiltonian is

$$\mathcal{H} = \frac{1}{4} \sum_{i=1}^{N-1} J_{i,i+1} \sigma_i^x \sigma_{i+1}^x + \frac{1}{2} \sum_{i=1}^{N} \Gamma_i^z \sigma_i^z, \tag{14}$$

with $\sigma_i$ the Pauli matrices for the spin $i$. This system is exactly solvable by Jordan-Wigner transformation [38], making it an ideal correctness checker for the DACP method. The nearest neighbor exchange interaction constants $J_{i,i+1}$ are random numbers that uniformly distributed in $[-J/\sqrt{N}, J/\sqrt{N}]$ with $J = 10$. The local random magnetic fields are represented by $\Gamma_i^z$, which are random numbers that uniformly distributed in the interval $[0, \Gamma]$ with $\Gamma = 1$.

Another system is the spin glass shards [27], which represents a class of global-range interacting systems that require relatively large bond dimensions to be tackled by the DMRG methods [41]. The Hamiltonian describing the system is

$$\mathcal{H} = \sum_{i<j} J_{ij} \sigma_i^x \sigma_j^x + \sum_i \Gamma_i^z \sigma_i^z. \tag{15}$$

All symbols and parameters are the same as that of the above Ising model, except that the first summation runs over all possible spin pairs and $J = 0.866\Gamma$. This system is interesting because it presents two crossovers from integrability to quantum chaos and back to integrability again. In the limit $J/\Gamma \to 0$, the ground state is paramagnetic with all spins in the local field direction and the system is integrable [27]. In the opposite limit $J/\Gamma \to \infty$, the ground state is spin glass and the system is also integrable since there are $N$ operators ($\sigma_i^x$) commuting with the Hamiltonian. A quantum chaos region exists between these two limits. $J = 10\Gamma$ is approximately the border from the quantum chaos to the integrable (the spin glass side) when $N = 20$, while $J = 0.866\Gamma$ indicates the system is in pure quantum chaos phase [27].

By employing the upper-bound-estimator, which costs little extra computation and bounds up the largest absolute eigenvalue $E_0$, one may estimate $E_{max} = E_0$ and $E_{min} = -E_0$ [42]. For this setting we have utilized the symmetry of the density of states (DOS), a bell-shape profile centered at zero, in the many-spin systems. Since we require $5,000$ central eigenvalues, we may set $n = 4,000$, corresponding to a dimension $8,001$ and being adequate to span the whole subspace $\mathbb{L}$. The overlap matrix $S$ is generally singular. The approximate distribution of DOS $\rho(E)$ may be efficiently calculated through the Fourier transformation of a time evolved wave function or through a better estimation method given in Ref. [43]. The parameter $a$ is appropriately chosen to ensure that the number of eigenstates contained in $[-a, a]$ is a little less than $8,000$ (as illustrated in Figure 3, the precision of some converged eigenvalues may be lower than required).

In practice, sometimes there may exist highly near-degenerate eigenvalues, with level spacings as small as $10^{-7}\Gamma$ while the average spacing is $10^{-5}\Gamma$. It is still hard (two magnitudes longer time) for the Chebyshev evolution to discriminate such close pair of eigenvalues. To circumvent this challenge, we employ the block filter technique [29], which means a block of states is filtered or evolved "simultaneously", in programming of the DACP method. The idea is that two or several random states in the degenerate subspace are usually linearly independent. For each numerical test, a block of 5 initial trial states is randomly generated and employed with the parameter $n$ being adjusted to $n = 800$ accordingly.

By these settings, we perform numerical tests on the above two systems to show the exactness and efficiency of DACP method. For this work, we consider only the eigenvalues computations. All the timing information reported in this manuscript is obtained from calculations on the Intel(R) Xeon(R) CPU E7-8880 v4, using sequential mode.

In Figure 3, we present the relative error $\eta$ in logarithmic scale versus the system energy $E_{exact}$, for the Ising model with $N = 19$ and the spin glass shards with $N = 17$. We have defined the relative error $\eta$ of the computed central eigenvalues $E$ as

$$\eta = \left| \frac{E - E_{exact}}{E_{exact}} \right|.$$

Exact eigenvalues of both systems have been obtained by other reliable methods. For the Ising model, we make use of the famous Jordan-Wigner transformation to reduce the original $2^N \times 2^N$

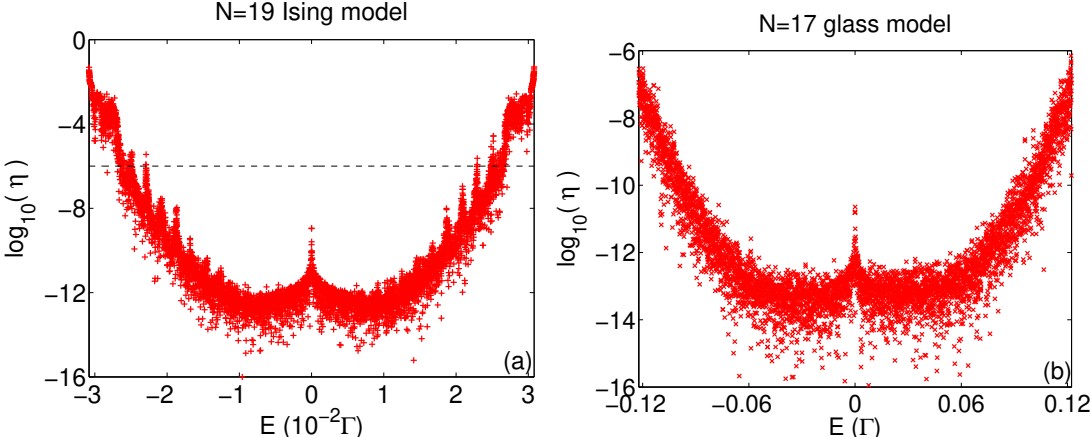

Figure 3: The relative error $\eta$ in logarithmic scale of the calculated eigenvalues, for (a) Ising model with $N = 19$ and (b) spin glass shards model with $N = 17$. The horizontal axis is the system energy. The number of eigenvalues satisfying $\eta < 10^{-6}$ (black dashed line) is $5,385$ for (a) and $5,000$ for (b). Such a distribution comes from the shape of filters in Figure 1.

matrix to a $2N \times 2N$ one, and restore the full spectrum of the original Hamiltonian [38]. For the spin glass shards we simply utilize the function $eigs$ of MATLAB, to find $5,000$ eigenvalues closest to $E = 0$. As for our numerical tests, the parameter $a$ in Figure 3 is $0.036\Gamma$ and $0.16\Gamma$ for (a) and (b), respectively. In computing the Ising model, the number of eigenvalues satisfying $\eta < 10^{-6}$ is not enough (less than $5,000$) by the settings mentioned above. By expanding the block size to 10 and the parameter $n$ to 500, we then collect enough eigenvalues. The number of converged eigenvalues, i.e., computed eigenvalues satisfying the condition $\eta < 10^{-6}$ is $5,385$ for (a) and $5,000$ (all the exact eigenvalues we have) for (b), while the total number of computed eigenvalues is $6,232$ and $5,910$, respectively.

The spike around $E = 0$ for both figures is due to the smallness of the denominator $E_{\text{exact}}$. The smallest absolute eigenvalue is about $4.4 \times 10^{-6}\Gamma$ for (a) and $2.9 \times 10^{-5}\Gamma$ for (b). Besides, there is a flat plateau at the middle of the figures, indicating that for those eigenvalues around $E = 0$ we encounter the numerical error, i.e., the absolute error reaches the limit of the double precision representation. In Figure 3(a) one may observe other spikes, which do not appear in (b). Each of those spikes corresponds to a cluster of extremely close eigenvalues. The integrability of the Ising model implies the level clustering and a Poissonian distribution of energy level spacings. We speculate that the Chebyshev evolution does not function well for the near-degenerate eigenvalues, as they give quite similar phase contributions to the final states. To significantly amplify the phase difference $\exp(i\Delta E t)$, it requires $t \gtrsim 1/\Delta E$, where $\Delta E$ is the energy difference. An alternative solution is to increase the block size to match up with the dimension of near-degenerate subspace. Ignoring the central plateau and the spikes, the distribution of $\eta$ shows an inverted shape of the exp-semicircle filter. We discuss their relationship in Appendix E.

In Figure 4, we further present the statistical distributions $P(s)$ of the level spacing $s$ (the difference between consecutive eigenvalues, and in units of mean spacing) for the two systems. The statistical distribution in Figure 4(a) is constructed from $5,000$ central eigenvalues (nearest to $E = 0$), while the distribution in Figure 4(b) is obtained for $2,500$ central eigenvalues of the same symmetry class (the interaction in Eq. (15) does not mix states with even and odd numbers of spins up). The transverse field Ising model is integrable, exhibiting a Poisson distribution $P(s) = e^{-s}$ of level spacings. On the other hand, for the spin glass shards model, we set $J = 0.866\Gamma$ in Eq. (15), corresponding to the quantum chaos phase [27]. Indeed, the

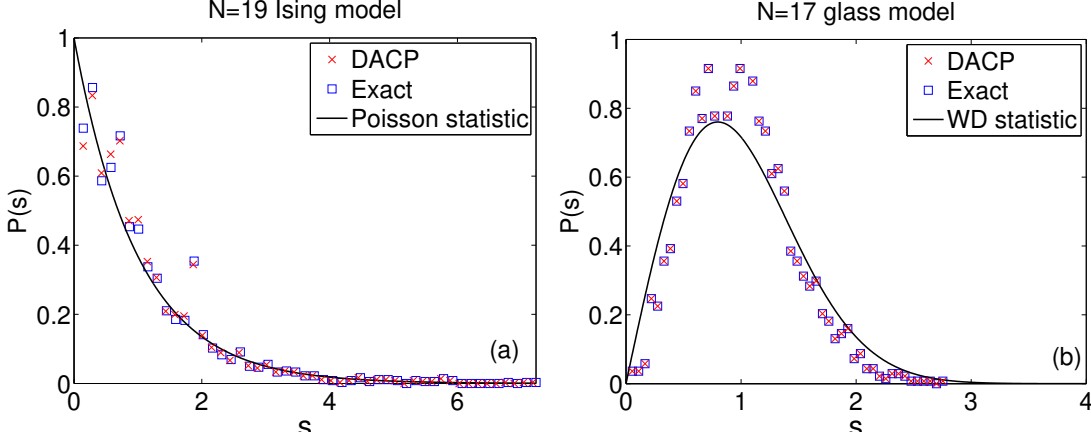

Figure 4: Statistical distributions of the level spacings $s$, for (a) Ising model with $N = 19$ and (b) spin glass shards model with $N = 18$. The probability distributions $P(s)$ come from the numerical results of DACP (red circles), the exact eigenvalues (blue squares), Poisson statistic $P(s) = \exp(-s)$ (black curve in (a)) and Wigner-Dyson GOE statistic (black curve in (b)). The horizontal axis $s$ is in units of mean level spacing.

distribution in Figure(b) obeys Wigner-Dyson Gaussian orthogonal ensemble (GOE) statistic.

In Figure 5, we show the computation time versus the number of converged eigenvalues, for the two systems of size $N = 15$. The horizontal axis represents the number of computed eigenvalues satisfying the condition $\eta(E) < 10^{-6}$ in each test. Recall that the DACP method is divided into three parts: the exp-semicircle filtration, Chebyshev evolution, and subspace diagonalization. We denote their time consumption as $T_F$, $T_E$, and $T_D$, respectively, while solving time is the full computation time $T = T_F + T_E + T_D$. One immediately notices that for the two systems, computation time $T$ decreases on the left side, reaching a plateau at the middle region, while quickly increases on the right side. Yet the dashed lines $T_E$ present a quite small increasing rate for the whole region, as the horizontal axis is in logarithmic scale. At the middle of the region, where the DACP method performs the best, the solving time $T$ and the evolution time $T_E$ roughly coincide with each other.

These features are easily understood if one considers the following four facts. (i) Notice that the DOS $\rho(E)$ is usually a bell-shape profile peaked at zero in spin systems and that $a$ is typically a tiny parameter ($a/E_{\max} \sim 10^{-2}$ for $N = 19$), thus one may safely take $\rho(E)$ as a constant $\bar{\rho}$ in $[-a, a]$. Consequently, the number of required eigenvalues is $R \simeq 2\bar{\rho}a$. (ii) Setting the action of the Hamiltonian on the state, $\mathcal{H}|\psi\rangle$, as a basic step, and denoting the computation time of the basic step as $\tau$, we may count the filtration time as $\tau$ times twice the cut-off order $K = 12E_{\max}/a$, i.e., $T_F \simeq 24E_{\max}\tau/a \propto R^{-1}$. (iii) Similarly, we find the Chebyshev evolution time is $T_E \simeq \lfloor R\pi E_{\max}/a\rfloor\tau \propto R^0$ since $R \propto a$. (iv) It is well known that the full diagonalization time is proportional to the cube of the matrix size, i.e., $T_D \propto R^3$. The combination of these computation times clearly explains the behavior shown in Figure 5. For the left side, the exp-semicircle filtration dominates, as $T_F$ is inversely proportional to the parameter $R$. Whereas, when it comes to the intermediate region where the number of eigenvalues is several hundreds to thousands, the evolution time $T_E$ consists of the majority of the computation time while the other two are negligible. As shown in Figure 5, the performance of DACP method is approximately the same in finding 100 to 3,000 eigenvalues. As $R$ keeps increasing, the subspace diagonalization time $T_D$ eventually consumes the most computation time.

We remark that the plateau at the middle region in Figure 5 distinguishes the DACP method

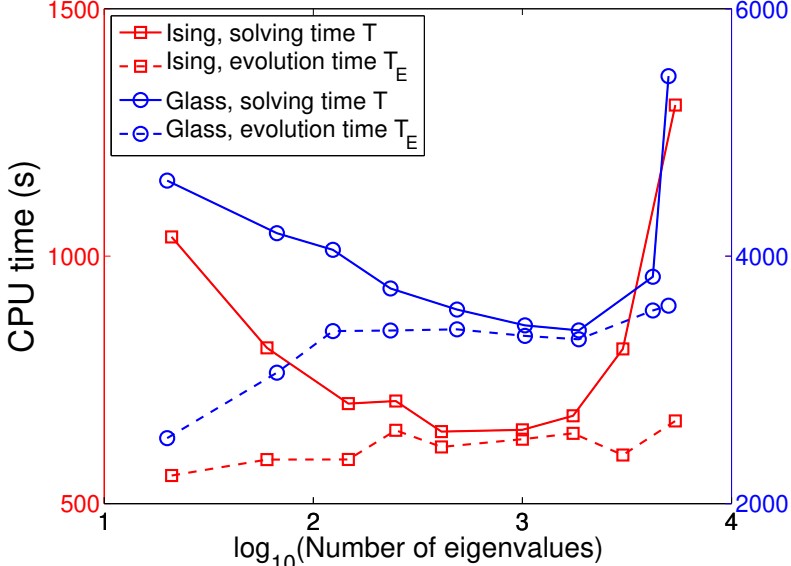

Figure 5: Computation time (CPU seconds) versus number of eigenvalues satisfying the condition $\eta < 10^{-6}$, for the Ising model (red lines with squares, left axis) and the spin glass shards (blue lines with circles, right axis), using the DACP method. For both systems, $N = 15$. The solid lines show the solving time $T = T_F + T_E + T_D$, including computation time of the exp-semicircle filtration $T_F$, of the Chebyshev evolution $T_E$ and of the subspace diagonalization $T_D$. The dashed lines present $T_E$.

from those iterative filtering ones [10, 20, 23, 24, 28, 29], although all of them make use of the Chebyshev polynomials. The iterative methods usually require a larger amount of filtrations and reorthogonalizations, thus a longer convergence time, in finding more eigenvalues [8, 20, 22]. In fact, as illustrated in Ref. [8], one may find that the computation time is roughly proportional to the number of eigenvalues required for the shift-invert approach. The DACP method is thus highly desirable for large scale eigenvalues computations.

At last, we note that the consumed memories by DACP method are rather small, as it works in a matrix-vector product mode which avoids an explicit matrix representation of the Hamiltonian. In addition, as shown in Appendix C, the whole set of states in Eq. (9) are not preserved in memory as well. The major memory consumption is the storage of elements $H_{ij}$ and $S_{ij}$, thus the memory requirements of the DACP method relate only to the dimension $d$ of subspace $\mathbb{L}$. By the settings of this work, it occupies around 5.6 GB of memory for appropriately calculating 5,000 eigenvalues in $N \leq 20$ systems.

## 4   Comparison with other approaches

We compare the performance of DACP method to other established methods in computing the two models introduced in Section 3. Specifically, we select three different types of methods listed and discussed as follows.

The shift-invert approach is widely used in computing eigenpairs at the middle of the spectrum for quantum spin systems, to name a few, like in [8, 15–17]. In our tests, it is implemented by the "*eigs*" function of Matlab R2019b, which employs the implicitly restarted Arnoldi method (ARPACK) [44]. The matrix of the Hamiltonian $\mathcal{H}$ is fully stored in memory, so one may expect its fast computation at the expense of high memory usage.

The Eigenvalues Slicing Library (EVSL) provides routines for computing eigenvalues lo-

cated in a given interval of real symmetric eigenvalue problems [21]. We choose the routine "*ChebLanTr*", which relies on polynomial filtering and is coupled with Krylov subspace method and the subspace iteration algorithm [32]. Since it allows a matrix-free format, we implement exactly the same matrix-vector product function as in the DACP method. However, in the EVSL the matrix-vector product mode (*SetAMatvec*) is restricted to apply to the real vectors, while the quantum states are generally complex ones. Therefore, all the computation times of the EVSL shown in the figures are twice that of recorded in practice.

The FEAST algorithm is a general purpose eigenvalue solver which takes its inspiration from the density-matrix representation and contour integration technique in quantum mechanics [45, 46]. In our tests, we have installed the FEAST package v4.0 and invoked the solver function "*zfeast_hcsrev*". Although the matrix-free mode is utilizable, it demands users to provide direct/iterative linear system solvers combined with the matrix-vector routines. For simplicity, we store the matrix of Hamiltonian in the sparse-CSR format. During the tests we find that it may not converge when the wanted number of eigenvalues is several thousands, we thus divide the original search interval into several smaller ones.

As for the DACP method, to better recognize its scaling behavior, we have discarded the time consumption of the subspace diagonalization, which is a constant that relies only on the size of the subspace. Note that simply adding or subtracting a constant does not affect the scaling. Specifically, the subspace diagonalization spends roughly 700 CPU seconds in each test, and such a constant is negligible for $N \geq 16$ systems. Nevertheless, we plot the full computation time of the DACP method in dashed lines.

Before the comparison of scaling behaviors, we find that both the EVSL and the FEAST

Table 1: Scaling constants $\alpha$ by linear fitting of the curves in Figure 6.

| $\alpha$ | DACP | MATLAB | EVSL | FEAST |
|---|---|---|---|---|
| Ising | 1.49 | 1.42 | 1.31 | 1.29 |
| Glass | 1.50 | 1.47 | 1.45 | 1.53 |

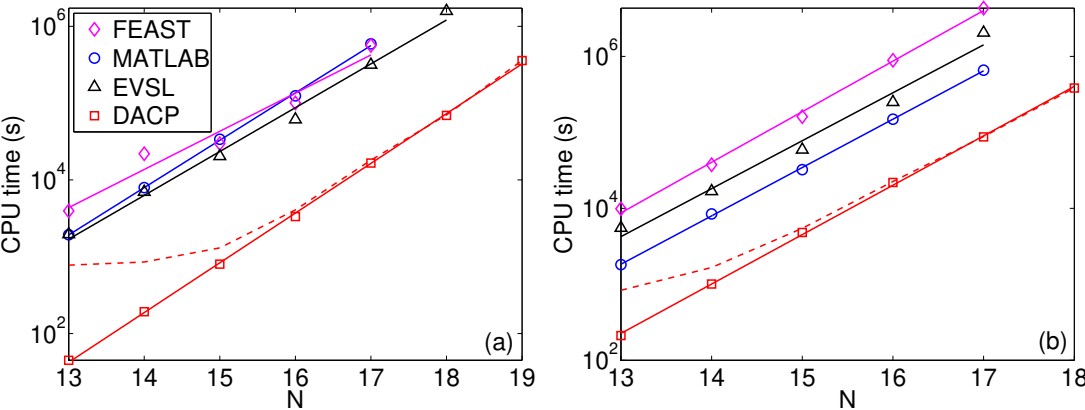

Figure 6: Scaling behavior measured by the computation time (CPU seconds) versus the system size $N$, for (a) Ising model and (b) spin glass shards model. In both panels, we compare the FEAST algorithm (pink diamonds), the shift-invert method (blue circles) and the EVSL (black triangles) with the DACP method (red squares). Each of the numerical tests (plotted as different marks) illustrates the computation time in finding $5,000$ central eigenvalues. The solid lines are obtained via linear fittings. For the DACP method, the red dashed lines present the full solving time $T$ while the red solid squares present the evolution time $T_E$.

approaches are not capable to find 5,000 eigenvalues in a single run (one search interval). Therefore, one must divide the search interval to achieve a better performance. We suppose that each smaller interval contains a similar amount of eigenvalues, where the number is determined by their computational performances.

In Figure 6, we compare the scaling behaviors measured by computation time $T$ (CPU time in seconds) of the DACP method with three different types of method, i.e., the shift-invert, the EVSL and the FEAST approaches. Each one of the numerical tests (marks) in Figure 6 essentially represents the computation time in finding 5,000 eigenvalues. During the tests, the parameter $M_0 = 1.2R$ for the FEAST. Considering the performance of the two methods and for the sake of convenience, we set the number of eigenvalues in each search interval as 500 for the EVSL and 1,000 for the FEAST. Both the DACP method and the shift-invert approach successfully find 5,000 eigenvalues in a single run. The EVSL approach also finds 5,000 eigenvalues by the "divide and conquer" technique. Whereas, due to its long computation time, we set up the FEAST to find 1,000 eigenvalues (a single interval) and the total computation time is 5 times that.

As shown in Figure 6, for spin systems with $13 \leq N \leq 17$, the DACP method is about 20 times faster than the other approaches for both models, except for comparing to MATLAB for the spin glass shards model. Since the shift-invert method essentially finds the ground energy of $\mathcal{H}^{-1}$, which is usually not a sparse matrix, its computation time $T$ is not affected by the sparsity of the Hamiltonian and the two blue lines in Figure 6(a) and (b) are nearly the same. On the contrary, since the DACP method employs the polynomial combination of $\mathcal{H}$ acting on the states, its computation time $T$ is heavily influenced by the number of Pauli operators of the specific Hamiltonian. For example, when $N = 18$ the computation time $T$ for the spin glass shards is 5.5 times that for the Ising model, while the number of Pauli operators is 6.2 times. Considering this effect, the DACP method is still advantageous over the shift-invert approach in the worst case where the exchange interactions run over all possible spin pairs and all three directions.

Furthermore, the scalings among these four methods are comparable. To compare quantitatively, we extract the scaling constants $\alpha$ by fitting the numerical results, where $\alpha$ is defined by $T = T_0 \exp(\alpha N)$. The values of $\alpha$ are shown in Table 1. Indeed, the scaling constants are quite close for the spin glass shards model, while both the EVSL and FEAST own smaller $\alpha$ for the Ising model. Suppose that $\alpha$ remains unchanged for a bigger $N$, then the DACP method's advantage in time cost may keep on for rather large systems. Specifically, as for the Ising model, the scaling lines between the FEAST and DACP cross at $N \simeq 28$, while for the glass model, the scaling lines between the EVSL and DACP cross at $N \simeq 80$. In addition, as illustrated by the numerical tests of the shift-invert method in Ref. [8], the factorization time for finding $\mathcal{H}^{-1}$ dominates other computation steps. Considering the factorization part only, the execution time in Ref. [8] exhibits a scaling constant $\alpha \simeq 1.66$, indicating a worse scaling behavior for systems with larger spins. The time efficiency of the DACP method is thus confirmed.

Finally, in Table 2 we compare the memory consumption of these approaches for the Ising systems with $N \geq 16$. As the subspace matrix elements constitutes the major storage of DACP

Table 2: Memory consumption of the four approaches for the Ising model.

| Memory (GB) | DACP | MATLAB | EVSL | FEAST |
|---|---|---|---|---|
| N=16 | 5.6 | 144.9 | 1.2 | 50.1 |
| N=17 | 5.6 | 573 | 2.9 | 99.6 |
| N=18 | 5.6 | \ | 3.2 | 395.5 |

method, its memory requirement remains a constant during the tests. More precisely, a quantum state of $N = 18$ systems occupies about 4 MB of memory, and there are only around 10 quantum states kept in the memory. The dominant term is the full storage of two matrices $S$ and $H$, which remains a constant as $N$ increases. On the contrary, for the shift-invert method, the matrix size of $\mathcal{H}^{-1}$ grows rather fast as the system size $N$ increases (proportional to $4^N$), demanding a large amount of memories. For example, in our tests it consumes 573 GB of memory for $N = 17$ systems, which is already a factor of 100 more than that of the DACP method. Since the EVSL approach also works in a matrix-free mode and since it calculates only 500 eigenvalues (subspace dimension $d \simeq 1,000$) in one run, it requires the least memory. Using the sparse matrix format, the FEAST approach saves a lot of memories while is still significantly more than the matrix-free ones.

## 5   Conclusion

We propose the DACP method to efficiently calculate a large scale (at least $5,000$) of central eigenvalues with high precision, by employing twice the Chebyshev polynomials. It explores several excellent properties of Chebyshev polynomial, to efficiently filter out unwanted amplitudes and to construct the appropriate basis states for the middle of the spectrum. In particular, the proposed method is specified to solve the middle of the spectrum (eigenvalues around $E = 0$). This restriction makes it possible to combine different properties of Chebyshev polynomial together. Eventually, two key features distinguish DACP from the polynomial filtering methods: its computation time depends weakly on the number of required eigenvalues, while its memory overhead is independent of the system size. Moreover, as shown in Appendix D, the DACP method is more stable and more efficient than the Lanczos method applied with the exp-semicircle filter.

The numerical tests for the Ising model and the spin glass shards confirm the exactness and efficiency of the DACP method. Compared to the widely used shift-invert approach, the DACP method gives a considerable increase in the speed of computations, for the Ising model up to a factor of 30 while for the spin glass shards the increase in speed is less but still considerable (a factor of 8). Besides, the memory requirements are drastically decreased, up to a factor of 100 for $N = 17$ spin systems and even more for larger ones. As a powerful tool for central eigenvalues calculations, the DACP method may find potential applications in many physical problems, such as many-body localization in condensed matter physics [7, 15, 17, 47], and level statistics in quantum chaos [1, 16, 27, 48].

## Acknowledgements

We thank X.-H. Deng for discussions. The numerical calculations in this paper have been done on the supercomputing system in the Supercomputing Center of Wuhan University.

**Funding information**   This work is supported by the National Natural Science Foundation of China (NSFC) Grant No. 91836101, No. U1930201, and No. 11574239.

## A   Chebyshev polynomials

In this paper, we limit ourselves to that only the polynomial combinations of $\mathcal{H}$ are the allowed operations. We utilize the Chebyshev polynomials to fulfill the tasks mentioned above. It is the

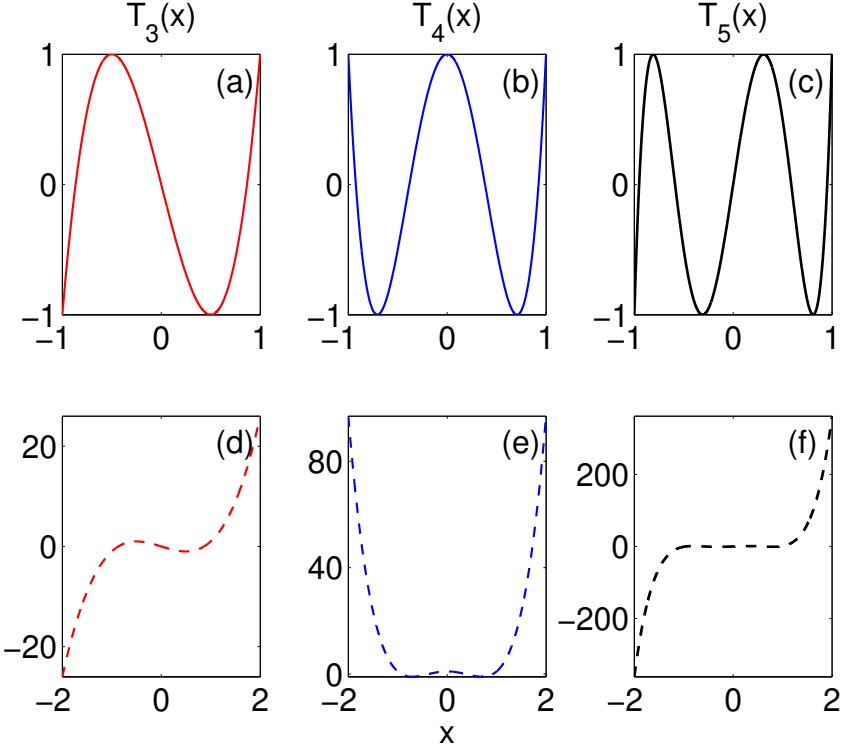

Figure 7: Chebyshev polynomials $T_k(x)$ for $k = 3$ (red lines), $k = 4$ (blue lines), $k = 5$ (black lines). The first row ((a-c) with solid lines) illustrates the oscillations of $T_k(x)$ inside the interval $[-1, 1]$ while the second row ((d-f) with dashed lines) shows the rapid increase outside $[-1, 1]$ of Chebyshev polynomials.

key to bridge the combinations of $\mathcal{H}^k$ with an approximately exponential function of $\mathcal{H}$, either $\exp(k\mathcal{H})$ or $\exp(ik\mathcal{H})$. Due to its several remarkable properties, the Chebyshev polynomials may exhaust the potential of this type of methods.

The $k$th order Chebyshev polynomial of the first kind is defined by

$$T_k(x) = \begin{cases} \cos\left(k\cos^{-1}(x)\right), & |x| \leq 1 \\ \cosh\left(k\cosh^{-1}(x)\right), & x > 1 \\ (-1)^k \cosh\left(k\cosh^{-1}(-x)\right), & x < -1 \end{cases}, \tag{A.1}$$

with initial conditions $T_0(x) = 1$ and $T_1(x) = x$ [49]. It is a piece-wise function containing two different kinds of expression, while being the polynomial function, it is both continuous and smooth. For simplicity, let us set $\theta = \cos^{-1}(x)$ ($\cos\theta = x$) when $x \in [-1, 1]$ and set $\theta = \cosh^{-1}(x)$ when $x \in [1, \infty)$, the corresponding range of $\theta$ is $\theta \in [0, \pi]$ and $\theta \in [0, \infty)$, respectively. In terms of the variable $\theta$, Eq. (A.1) becomes

$$T_k(x) = \begin{cases} \cos(k\theta), & |x| \leq 1, \\ \cosh(k\theta), & x > 1. \end{cases} \tag{A.2}$$

One may easily observe that $T_k(x)$ is a sine or cosine-like oscillation function bounded by $-1$ and $1$ inside the interval $[-1, 1]$, as illustrated in Figure 7(a-c), while it grows extremely fast outside $[-1, 1]$, as shown in Figure 7(d-f).

Note that $\cosh(k\theta) = (e^{k\theta} + e^{-k\theta})/2$. It is natural to expect an exponential growth of the Chebyshev polynomial outside the interval $[-1, 1]$. In fact, it is known that among all polynomials with degree $\leq k$, the Chebyshev polynomial $T_k(x)$ grows the fastest outside the interval $[-1, 1]$ under comparable conditions [31].

Associated with those properties is a practically useful one: $T_{k+1}(x)$ can be efficiently determined by the 3-term recurrence

$$T_{k+1}(x) = 2x T_k(x) - T_{k-1}(x). \tag{A.3}$$

All these properties of the Chebyshev polynomial render it a powerful toolbox and are of great use for the DACP method.

## B  Detailed deduction of Eq. (9)

Here we derive explicit expressions for constructing the set Eq. (9) via the Chebyshev evolution. We focus on the case that $a \ll E_{\max}$, which is fairly reasonable for large ($N \geq 15$) spin systems.

$$
\begin{aligned}
|\psi_E(k)\rangle &= T_k(\mathcal{G}) |\psi_E\rangle \\
&= \sum_j \cos(k\omega_j) c'_j |\phi_j\rangle \\
&\simeq \sum_j \cos(\frac{k\pi}{2} - \frac{kE_j}{E_{\max}}) c'_j |\phi_j\rangle \\
&= \begin{cases} (-1)^n \sum_j \cos(k\frac{E_j}{E_{\max}}) c'_j |\phi_j\rangle, & k = 2n, \\[2mm] (-1)^n \sum_j \sin(k\frac{E_j}{E_{\max}}) c'_j |\phi_j\rangle, & k = 2n+1. \end{cases}
\end{aligned}
\tag{B.1}
$$

Using the fact that when $x$ is small, $\arccos(x) = \pi/2 - x + o(x)$, we obtain $\omega_j = \arccos(E_j/E_{\max}) \simeq \pi/2 - E_j/E_{\max}$ at the third line of Eq. (B.1). Therefore, we find the expression

$$
T_k(\mathcal{G}) \simeq \begin{cases} (-1)^n \cos(k\mathcal{G}), & k = 2n, \\ (-1)^n \sin(k\mathcal{G}), & k = 2n+1. \end{cases}
\tag{B.2}
$$

We then conduct a Chebyshev evolution with a cutoff order $K' = \lfloor n\pi/a_r \rfloor$, recording both $T_{k-1}(\mathcal{G})|\psi_E\rangle$ and $T_k(\mathcal{G})|\psi_E\rangle$ when $k = \lfloor m\pi/a_r \rfloor$ with $m = 1, \cdots, n$. After the evolution, the set of states Eq. (9) is automatically generated.

## C  Evaluation of matrix elements and solution of the generalized eigenvalue problem

As shown in Appendix B, we rewrite the basis Eq. (9), or $\{|\Psi_i\rangle : i = 1, \cdots, 2n+1\}$, using the Chebyshev polynomials:

$$
\left\{ I, T_{k_1-1}(\mathcal{G}), T_{k_1}(\mathcal{G}) \cdots, T_{k_n-1}(\mathcal{G}), T_{k_n}(\mathcal{G}) \right\} |\psi_E\rangle,
\tag{C.3}
$$

where $k_m = \lfloor m\pi/a_r \rfloor$, $m = 1, \cdots, n$, $|\Psi_1\rangle = |\psi_E\rangle$, $|\Psi_2\rangle = T_{k_1-1}(\mathcal{G})|\psi_E\rangle$, $|\Psi_3\rangle = T_{k_1}(\mathcal{G})|\psi_E\rangle$, etc. For simplicity, one may further assume $k_m$ is an even number.

The elements $S_{ij} = \langle \Psi_i | \Psi_j \rangle = \langle \psi_E | T_x(\mathcal{G}) T_y(\mathcal{G}) | \psi_E \rangle$, where $x$ and $y$ are directly determined by $i$ and $j$, respectively:

$$
x = \begin{cases} k_{i/2} - 1, & \text{for even } i, \\ k_{(i-1)/2}, & \text{for odd } i, \end{cases}
$$

while $y$ and $j$ share the same relation. By making use of the relation [49]

$$T_i(\mathcal{H})T_j(\mathcal{H}) = \frac{1}{2}(T_{i+j}(\mathcal{H}) + T_{|i-j|}(\mathcal{H})), \tag{C.4}$$

one may even find the matrix elements without recording any states during the Chebyshev evolution. Instead, we simply record the two numbers $\langle\psi_E|T_k(\mathcal{G})|\psi_E\rangle$ and $\langle\psi_E|\mathcal{H}T_k(\mathcal{G})|\psi_E\rangle$ at the appropriate time, i.e., when $k = k_m - 2$, $k = k_m - 1$, $k = k_m$ and $k = k_m + 1$.

Finally, we arrive at the explicit expressions of matrix elements

$$S_{ij} = \frac{1}{2} \langle\psi_E|\left(T_{x+y}(\mathcal{G}) + T_{|x-y|}(\mathcal{G})\right)|\psi_E\rangle, \tag{C.5}$$

$$H_{ij} = \frac{1}{2} \langle\psi_E|\mathcal{H}\left(T_{x+y}(\mathcal{G}) + T_{|x-y|}(\mathcal{G})\right)|\psi_E\rangle. \tag{C.6}$$

Also note that since $T_{x+y}$ is needed, where both $x$ and $y$ may reach the maximum value $k_n$, the cut-off order $K$ is doubled to $K = 2k_n$ in this mode. When the block technique is employed, the situation becomes a little more complicated. One then needs to record the cross terms (like a $2 \times 2$ matrix) instead of the single number. In this case, the relationship between $i, j$ and $x, y$ remains the same. Suppose we have two different evolving states $|\psi_1\rangle$ and $|\psi_2\rangle$, the three numbers

$$\langle\psi_1|T_k(\mathcal{G})|\psi_1\rangle, \langle\psi_1|T_k(\mathcal{G})|\psi_2\rangle \text{ and } \langle\psi_2|T_k(\mathcal{G})|\psi_2\rangle$$

shall be recorded at the same time (same $k$). The final version of $S$ matrix is then composed of 4 matrices:

$$S = \begin{bmatrix} S^{11} & S^{12} \\ S^{21} & S^{22} \end{bmatrix}, \tag{C.7}$$

where $S_{ij}^{ab} = \langle\psi_a|T_x(\mathcal{G})T_y(\mathcal{G})|\psi_b\rangle$.

For the generalized eigenvalue problem Eq. (11), the Hermitian matrix $S$ is first diagonalized as

$$S = V\Lambda_s V^\dagger, \tag{C.8}$$

where $V$ is the eigenvector matrix for $S$, $VV^\dagger = I$, and $\Lambda_s$ is the associated eigenvalue matrix. Since $S$ is generally singular, we then contract the $(2n+1) \times (2n+1)$ matrix $V$ by elimination of the columns associated with eigenvalues with absolute value below a cutoff $\varepsilon = 10^{-12}$. Denoting the number of retained eigenvalues as $m$, the contracted eigenvector matrix $\widetilde{V}$ is of order $(2n+1) \times m$, and

$$\widetilde{S} = \widetilde{V}\widetilde{\Lambda}_s\widetilde{V}^\dagger. \tag{C.9}$$

The next step is to form the contracted Hamiltonian matrix $\widetilde{H}$. Since

$$I = \left(\widetilde{\Lambda}_s^{-\frac{1}{2}}\widetilde{V}^\dagger\right)\widetilde{S}\left(\widetilde{V}\widetilde{\Lambda}_s^{-\frac{1}{2}}\right), \tag{C.10}$$

denoting the transformation matrix $U = \widetilde{V}\widetilde{\Lambda}_s^{-\frac{1}{2}}$, the contracted $m \times m$ Hamiltonian matrix is

$$\widetilde{H} = U^\dagger H U. \tag{C.11}$$

The Hermitian matrix $\widetilde{H}$ of order $m \times m$ with $m$ ranging from $10^3$ to $10^4$ is then diagonalized directly

$$\widetilde{H} = \widetilde{Y}\widetilde{\Lambda}\widetilde{Y}^\dagger, \tag{C.12}$$

where $\widetilde{Y}$ is the eigenvector matrix of $\widetilde{H}$ and $\widetilde{\Lambda}$ is a diagonal matrix consisting of those desired eigenvalues of the original Hamiltonian $\mathcal{H}$ contained in $[-a, a]$. The eigenstates of the projected Hamiltonian $H$ may be obtained through elementary matrix algebra:

$$B = U\widetilde{Y} = \widetilde{V}\widetilde{\Lambda}_s^{-\frac{1}{2}}\widetilde{Y}. \tag{C.13}$$

Denoting the Eq. (C.3) as a $2^N \times (2n+1)$ matrix $A$ with $|\chi_i\rangle$ being the $i$-th column, the eigenstates of the original Hamiltonian $\mathcal{H}$ contained in $[-a, a]$ is

$$\Phi = AB = A\widetilde{V}\widetilde{\Lambda}_s^{-\frac{1}{2}}\widetilde{Y}. \tag{C.14}$$

We finally get the eigenvalues $\widetilde{\Lambda}$ and eigenstates $\Phi$.

# D  Comparison with a hybrid method

Since the Lanczos method is widely employed to compute the lowest (or highest) eigenvalues, while the exp-semicircle filter provides a means to transform the central eigenvalues to the highest ones, naturally one may combine them together to efficiently compute the central eigenvalues, a way far simpler than the DACP method. But there are several reasons to prefer complicated techniques used in the DACP method to the standard construction of the Krylov space, especially when large scale computations are required.

First, it is known that the Lanczos method is not numerically stable under practical conditions. Specifically, although the Lanczos algorithm shows perfect properties in theory, in practice it loses many of its designed features, e.g., global orthogonality and linear independence among Lanczos recursion states [35,50,51]. These defects effectively limits the number of eigenvalues which can be computed. In addition, it seems that the emergence of generalized eigenvalue problem (to deal with non-orthogonal base states) is unavoidable due to the error accumulations [52,53]. The Chebyshev recursion, on the other hand, possesses many interesting properties common in both the ideal and practical calculations [50]. In particular, it is accurate and stable for $x \in [-1, 1]$, allowing the propagation in the Chebyshev order domain for tens of thousands of steps without accumulating significant errors [33].

Second, taking the reorthogonalization step into consideration, the efficiency of the Lanczos algorithm decreases. The reorthogonalization step is a necessary part in both the Lanczos and the Arnoldi method. When there is a large amount (e.g., several thousands) of eigenvalues to be computed, the total cost is actually dominated by the reorthogonalization [28,29,54]. Suppose the total required number of eigenvalues is $R$ and the Hilbert space dimension is $D$, then the Lanczos-type methods scale as $DR^2$, as every generated Ritz vector needs to be reorthogonalized against the existing ones (see also the discussions in Refs. [29,44]). In comparison, we have shown in Section 3 that $T_F \propto DR^{-1}$, $T_E \propto DR^0$, and $T_D \propto D^0R^3$, where each one of them is better than $DR^2$ ($R \ll D$). Although partial reorthogonalization schemes have been proposed, they result in an increased cost in computations as well as memory traffic [20,55], and they are not guaranteed to succeed when the accuracy requirement becomes strict [55]. Moreover, the DACP method and the Lanczos algorithm are different in terms of space complexity, as the former shows $\max(DR^0, R^2)$, while the latter requires $DR$ once the reorthogonalizations are needed [44]. Therefore, the DACP method has superiority over the Lanczos algorithm in both the time and the space complexity.

Finally, ignoring the requirement of reorthogonalizations, we find that the two methods are comparable in both the time and the space complexity. Recall that the Chebyshev evolution time, being the dominating term in the DACP method, becomes $T_E \simeq \pi RE_{\max}\tau/a$, where $\tau$ is the time for matrix-vector product. We then discuss the time complexity of a combined version, the Lanczos method with the exp-semicircle filter. As shown in Fig. 1 of the Supplemental Material in Ref. [22], and the discussions in Ref. [28], the number of Lanczos steps $m$ shall be at least twice the number of requested eigenvalues $R$, i.e., $m \geq 2R$. Since $2R$ iterations are needed, and every iteration requires a filter application with $T_F \simeq 24E_{\max}\tau/a$, the computation time reads $2R \cdot 24E_{\max}\tau/a = 48RE_{\max}\tau/a$. Therefore, the DACP method and the combination of Lanczos with the exp-semicircle filter share the same time complexity.

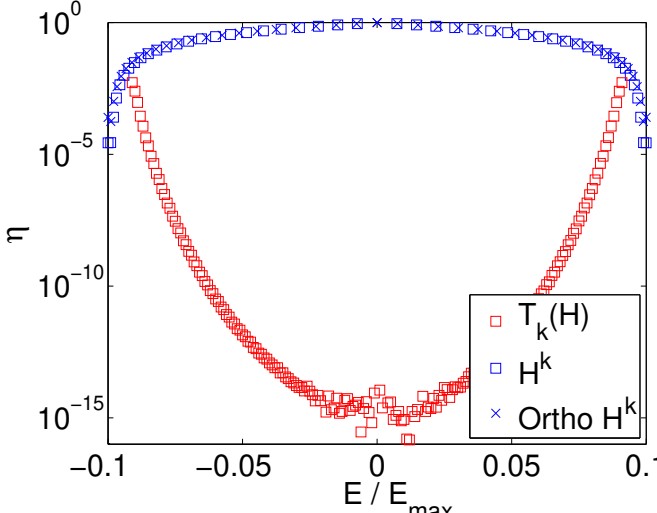

Figure 8: Comparison of relative error distributions $\eta(E)$ for different basis during the second step of the DACP method, including $\{T_k(\mathcal{H})\}$ (red squares), $\{\mathcal{H}^k\}$ (blue squares) and $\{\mathcal{H}^k\}$ with Schmidt orthogonalization (blue crosses).

In addition, we also test the basis set in Eq. (10) using the simulation of DACP (calculating the error produced by the three steps of DACP as the exact eigenvalues are known), with other conditions the same. The results are shown in Figure 8. Such a basis set does not converge to any eigenvalues in the condition $\eta(E) \leq 10^{-6}$. In contrast to the DACP method, the best converged eigenvalues for basis $\{\mathcal{H}^k\}$ are located at the two ends of the interval. In summary, the DACP method is more stable and more efficient than the Lanczos method applied with the exp-semicircle filter.

## E  Error analysis

In this section, we deduce the shape of relative error distribution. We obtain the error results using two different approaches. The first way roughly follows the progress of the DACP method except substituting those computationally hard steps by their direct results. The second one is analytical deductions under several approximations.

We first state the assumptions shared by the two approaches.

1. We start from an initial state $|\psi\rangle = \beta \sum_i c_i |E_i\rangle$, where $\beta$ is the normalization constant, $c_i$ the real valued probability amplitudes, $|E_i\rangle$ the eigenstates with $E_i$, and $-a \leq E_i \leq a$. Namely, $c_i = 0$ when $|E_i| > a$.

2. The exact energy levels are known and are equally spaced, i.e., $E_{i+1} - E_i = E_i - E_{i-1}$.

3. The maximum abstract energy $E_{max} = 1$.

4. We compute the eigenvalues following the Chebyshev evolution and the subspace diagonalization of the DACP method.

Under these settings, we employ a simulation program of the DACP method (representing partially analytical results) to efficiently generate errors that are close to that of the DACP method. In detail, instead of performing $K$ order Chebyshev evolution in the DACP method, we explicitly write down the expressions of the states in Eq. (9) and construct the corresponding subspace matrices in Eq. (12). The results are almost exactly the same to DACP, except

that there exists a slight difference between the two sides of Eq. (B.2). Nevertheless, such a simulation catches the major error produced during the Chebyshev evolution and subspace diagonalization without performing difficult computational tasks (assuming the exact eigenvalues are known). Moreover, it is also convenient to test the effectiveness of the basis states constructed from Eq. (10).

Next, we introduce the analytical deduction approach. The difference between the simulation approach and analytical one lies essentially in the third step of the DACP. The former one performs the full diagonalization of the subspace matrices. For simplicity while catching the essential, here we consider only two different eigenvalues $\{E_1 = 0, E_2\}$ combined with two constructed states $\{|\psi_1\rangle, |\psi_2\rangle = \sin(\pi\mathcal{H})|\psi_1\rangle\}$, where

$$|\psi_1\rangle = \beta_1(|E_1\rangle + \varepsilon |E_2\rangle) ,$$

$\varepsilon = c_2/c_1 \ll 1$, and $\beta_1 \simeq 1$. In the case of the distribution in Eq. (5), $c_1 \simeq \exp(24)$. $\varepsilon \ll 1$ means that $E_2$ is far away from $E_1 = 0$. Explicitly expressing $|\psi_2\rangle$, we then obtain

$$|\psi_2\rangle \simeq \varepsilon \sin(\pi E_2)|E_2\rangle .$$

The numerical round-off error (marked as $\delta$) runs in during the normalization process, namely, a small number like $\varepsilon^2$ is modified to $\varepsilon^2 + \delta$. Specifically, the expression of overlap matrix $S$ reads

$$S = \begin{bmatrix} 1 & \beta_1\varepsilon^2 \sin(\pi E_2) \\ \beta_1\varepsilon^2 \sin(\pi E_2) & \varepsilon^2 \sin^2(\pi E_2) \end{bmatrix}. \tag{E.1}$$

Supposing that $10^{-12} \le \varepsilon^2 \ll 1$, diagonalization of Eq. (E.1) gives

$$\Lambda_s \simeq \begin{bmatrix} 1 & 0 \\ 0 & \varepsilon^2 \sin^2(\pi E_2) \end{bmatrix}.$$

Correspondingly, the eigenvalues of the matrix $H$ in Eq. (C.11) are $\{E_1, E_2\varepsilon^2 \sin^2(\pi E_2)\}$, which are then multiplied by $\Lambda_s^{-\frac{1}{2}}$ twice (recall that $\widetilde{H} = \widetilde{\Lambda}_s^{-\frac{1}{2}} \widetilde{V}^\dagger H \widetilde{V} \widetilde{\Lambda}_s^{-\frac{1}{2}}$). Suffering from the round-off errors, the smallest number $E_2\varepsilon^2 \sin^2(\pi E_2)$ turns into $E_2\varepsilon^2 \sin^2(\pi E_2) + \delta$, generating a relative error

$$\eta \simeq \frac{\delta}{E_2\varepsilon^2} = \frac{\delta c_1^2}{E_2 c_2^2}. \tag{E.2}$$

Generalizing the Eq. (E.2) and using $c_1^2 \simeq \exp(48)$, one finally gets

$$\eta(E_i) \simeq \frac{\delta \exp(48)}{E_i c_i^2} \simeq \frac{2.8 \times 10^{-18}}{\beta^2 E_i c_i^2}. \tag{E.3}$$

In deducing the right hand side of Eq. (E.3), we have used that the double precision numbers have 16 significant digits and $\beta^2 \simeq 4 \times 10^{-23}$ for our simulation in Figure 9(a), where $\beta$ is the normalization constant that obeys $\beta^2 \sum_i c_i^2 = 1$. When the distribution of the probability amplitudes varies, $\beta$ is changed accordingly, giving different expressions of $\eta(E)$.

Alternatively, we may also consider a more complicated case, as the constructed set of states is $\{|\psi_1\rangle, |\psi_2\rangle = \cos(\pi\mathcal{H})|\psi_1\rangle\}$. We thus have

$$|\psi_1\rangle = \beta_1(|E_1\rangle + \varepsilon |E_2\rangle) ,$$
$$|\psi_2\rangle = \beta_1(|E_1\rangle + \varepsilon \cos(\pi E_2)|E_2\rangle) .$$

The matrix

$$S \simeq \begin{bmatrix} 1 & \frac{1+\varepsilon^2 \cos(\pi E_2)}{1+\varepsilon^2} \\ \frac{1+\varepsilon^2 \cos(\pi E_2)}{1+\varepsilon^2} & \frac{1+\varepsilon^2 \cos^2(\pi E_2)}{1+\varepsilon^2} \end{bmatrix}, \tag{E.4}$$

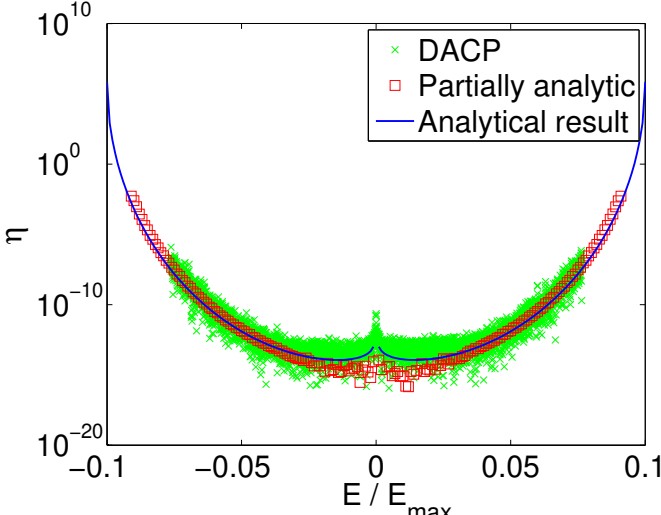

Figure 9: Comparison of relative error distributions $\eta(E)$ obtained via the DACP method (green crosses), simulation (partially analytical results) of DACP (red squares) and analytical prediction from Eq. (E.3) (blue curve) for the initial state $|\psi_E\rangle = \beta \sum_i c_i |E_i\rangle$. The results of DACP are the same as in Figure 3(b), except that the range is rescaled from $a = 0.16$ to $a = 0.1$. Note that in Eq. (E.3), the relative error $\eta(E)$ is proportional to $c_i^{-2}$, especially for those eigenvalues far away from $E = 0$.

gives the diagonal matrix

$$\Lambda_s \simeq \begin{bmatrix} 2 & 0 \\ 0 & \varepsilon^2 (1 - \cos(\pi E_2))^2 / 2 \end{bmatrix}.$$

A similar argument leads also to Eq. (E.3).

In Figure 9, we compare the relative error distributions of the above two approaches. Specifically, the results of the DACP method (the same as in Figure 3(b)) are also shown. The initial state for the former two is set directly to $|\psi\rangle = \beta \sum_i c_i |\phi_i\rangle$, where

$$c_i = \begin{cases} e^{\frac{2k}{E_{\max}} \sqrt{a^2 - E_i^2}}, & E_i \in [-a, a], \\ 0, & E_i \notin [-a, a]. \end{cases}$$

A similar state is prepared after the exp-semicircle filtering in the DACP method. Obviously, the distribution of $c_i$ is directly connected to the relative error $\eta(E)$, especially for those eigenvalues far away from $E = 0$. Therefore, to reduce error and to find more eigenvalues, the best initial state $|\psi_E\rangle$ for the Chebyshev evolution shall be the windows function

$$c_i = \begin{cases} \dfrac{1}{\sqrt{R}}, & E_i \in [-a, a], \\ 0, & E_i \notin [-a, a], \end{cases}$$

where $R$ is the dimension of eigenspace in $[-a, a]$.

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
