# Peer review of "Dual applications of Chebyshev polynomials method: Efficiently finding thousands of central eigenvalues for many-spin systems"

_SciPost Physics, doi:SciPost Phys. 11, 103 (2021)_

## Round 1 · Referee Report · Anonymous (Referee 1) · 2021-7-21

Strengths

  • The three-step method is simple and seems effective (runtime and memory)
  • A particularly nice part is the use of an exponential semicircle filter, which is probably optimal computationally (though this is not proven)
  • It is claimed that the creation of the non-orthogonal basis T_k(H)|psi_E> is more stable than the Lanczos scheme

Weaknesses

  • There is insufficient context given in the intro. The field of filtered interior eigensolvers is quite large. A comparison to other approaches like FEAST, EVSL or FILTLAN would be necessary.
  • What advantage does the proposed method bring over EVSL in particular?
  • It is unclear to me how a potentially ill-conditioned overlap matrix S is handled.

Report

This work could potentially meet the acceptance criteria ("breakthrough computational method"). This essentially hinges on whether the DACP method has some tangible advantages over more established methods.

Unfortunately this has not been shown convincingly in the paper. Only a comparison to shift-and-invert methods are mentioned. However there is a vast body of work on filtered interior eigensolvers in the literature. Two examples are FEAST (with a different approach) and EVSL (with a rather similar approach). I would recommend focusing the intro on a comparison to these methods, giving actual figures for the computational advantages of DACP (runtime, memory and/or stability).

My guess is that the strongest feature of the DACP approach compared to others is the filtering step, which is probably better (optimal?) compared to rational and polynomial filters, although this is not demonstrated. The second step (building the basis) is claimed to be better than Lanczos, but at first sight you pay the price of a potentially ill-conditioned overlap matrix S. How is this problem remedied? How precisely does an iteration with Chebyshev polynomials instead of H^k help in building the basis?

Requested changes

  • Devote a full section to comparing the DACP method to (a) simple Kernel Polynomial Method (eigenvalues from DOS), (b) EVSL and (c) FEAST approaches. Probably (a) is quite bad, but it is the most well known technique. No need for a detailed dissection/discussion of these methods, just runtime comparisons for the same problem (which will be undoubtedly in the interest of both the authors and the readers)

  • Discuss in a little more detail why the basis construction with T_k(H) is better than Lanczos. In particular, add to Appendix D a plot of the final eigenvalue errors and runtime using T_k(H) and H^k with and without reorthogonalization.

  • validity: high
  • significance: good
  • originality: good
  • clarity: high
  • formatting: good
  • grammar: reasonable

Author:  Haoyu Guan  on 2021-09-23  [id 1777]

(in reply to Report 1 on 2021-07-21)
Category:
answer to question

We appreciate the referee for the constructive suggestions, for recognizing the advantage of the exp-semicircle filter and for the positive assessment. We have revised the manuscript according to suggestions by both referees. Below, we address the questions and reply to them one by one.

1. There is insufficient context given in the intro.
We have added discussions on other filtered interior eigensolvers, and stated that instead of iterative filtering, the filtration in DACP is done only once.

2. The filtering step is probably better (optimal?) compared to rational and polynomial filters.
In the revised manuscript, we add Figure 2 to show that the exp-semicircle filter outperforms the other two polynomial filters in terms of the amplification factor. The fast growth of the Chebyshev polynomial in interval [1, 1+epsilon] is really impressive.

3. Pay the price of a potentially ill-conditioned overlap matrix S.
Indeed, we pay the price of a potentially ill-conditioned matrix S. The eigenvalues of S smaller than 1E-12 are discarded, leading to the contracted matrix \tilde{S}. In fact, this is essentially an orthogonalization process that is done in a low dimensional subspace. It is known as a standard procedure to avoid direct reorthogonalization, for example, see https://doi.org/10.1016/S0010-4655(98)00179-9 .

4. Compare the basis of Chebyshev polynomials with H^k.
We add a discussion on two error sources denoted as Appendix E in the revised manuscript. As the calculations show, a steep slope of the basis function in an extremely small interval is invaluable to separate the near-degenerate eigenvalues apart and to diminish the errors. In addition, we have tested the basis of H^k in the simulation program following the same procedures of DACP. The results are shown in Figure 8, Appendix D. We guess that the shape of error distribution in this case is dominated by the hardness to distinguish closely lied eigenvalues, which requires violently oscillating basis functions.

5. Devote a full section to compare the DACP method with others.
We add Section 4 in the revised version, to compare DACP with EVSL, FEAST and shift-invert methods in terms of both runtime and memory. The kernel polynomial method would be helpful in plotting the DOS and in deciding the value of parameter a (the length of target interval). But the precision of such a method is quite bad and it is not suitable to reveal the level spacing statistics. We also add Figure 4 to show that our results are capable to illustrate the typical spacing distributions.

---

## Round 1 · Referee Report · Anonymous (Referee 2) · 2021-7-27

Strengths

  • interesting topic
  • well written & easy to understand

Weaknesses

  • design choices for the algorithm not clearly motivated
  • no mathematical error analysis

Report

The authors propose a new numerical method to calculate interior eigenvalues of large sparse matrices, in particular eigenvalues of the Hamiltonians of two disordered interacting spin systems. This problem is relevant, for instance, to study the level-spacing distribution of the eigenvalues which characterizes quantum chaos and integrability. The method is based on a two-fold application of Chebyshev expansion: 1) as a spectral filter, and 2) to construct a basis of a restricted Hilbert space. Within this space eigenvalues are calculated using standard libraries.

Overall, the approach is plausible and seems to outperform other methods. However, I would prefer a more detailed explanation of the chosen design. Chebyshev expansion can be used for many other types of filters. Why did the authors pick the exp-semicircle filter? What is the particular advantage of this filter?

Figure 2 shows errors of the computed eigenvalues and attributes the particular shape of the error curve to the shape of the filter. Is there any theory for this observation, or mathematical error estimates? The authors also speculate that densely clustered eigenvalues reduce precision and cause the spikes in Figure 2(a). Is there any theory to describe this observation, or ideas to improve the algorithm?

The study of level-spacing statistics is the main motivation for calculating interior eigenvalues. The authors should provide examples of level-spacing distributions obtained with their method. Is the number of $10^3$ to $10^4$ eigenvalues calculated in one run sufficient for good statistics, or are multiple runs required (e.g., with different window position or different disorder sample)?

On the whole, the manuscript is well written with only few typos (e.g. 'midlle' on page 4). If the above remarks are taken into account it could be suitable for publication.

Concerning supplemental material: I was able to compile the C code provided by the authors, and it seemed to work. However, if readers are to benefit from the code it needs more comments, better structure and some simplification. Today such algorithmic examples can be presented very well with interactive Jupyter notebooks written in some free language like Python or Julia.

Requested changes

See report above. In short: - Explain the motivation for the exp-semicircle filter. What is the particular advantage of this filter? - Provide error estimates and try to explain the error data in Figure 2 with some math. - Show some data for level-spacing statistics calculated with DACP. - Fix typos. - If the C source code is to be attached to the paper, please add more comments and simplify. Maybe a Jupyter Notebook with descriptions and simple Julia or Python code is more instructive for the readers.

  • validity: high
  • significance: good
  • originality: good
  • clarity: high
  • formatting: excellent
  • grammar: excellent

Author:  Haoyu Guan  on 2021-09-23  [id 1778]

(in reply to Report 2 on 2021-07-27)

We thank the referee for constructive comments, useful advices, and for positively considering the manuscript could be suitable for publication. We have revised the manuscript according to suggestions by both referees. Below, we list the comments on the requested changes.

1. To illustrate the advantage of the exp-semicircle filter, we compare three different polynomial filters in Figure 2 of the revised manuscript. Indeed, Chebyshev expansion has been widely applied to other types of filters, especially the delta filter. Apparently, the exp-semicircle enormously outperforms the other two filters in comparing the amplification of probability amplitudes under a same number of matrix-vector products. This is due to the fastest growth property of the Chebyshev polynomials in the interval [1, 1+epsilon], which has not been explored by other filters.

2. We add Appendix E to discuss the major error source and to estimate the mathematical expression of error distributions, with the visualized results shown in Figure 9. The three distributions agree fairly well with each other.

3. We add Figure 4 in the revised manuscript to compare the level-spacing statistics of the results calculated with a single run of DACP and the Poissonian or Wigner-Dyson statistics. The numerical results and the analytical ones agree qualitatively.

4. We have examined and fixed typos.

5. We have revised and added more comments in the C source code. We shall learn to use interactive Jupyter notebooks and remark our code in the future.

---

## Round 2 · Referee Report · Pablo San-Jose (Referee 3) · 2021-10-14

Strengths
Same as before, plus: - A direct comparison to state of the art filtering and iterative methods shows very favorable performance of DACP - Very detailed analysis in version 2 of performance and convergence behaviors of the algorithm
Weaknesses
Can only find three minor weaknesses that could be improved: - Can the dimensions of the filtered subspace be known or estimated in advance? - The final diagonalization step can probably be accelerated by using QZ (generalized Schur) factorizations, instead of SVD of the overlap matrix followed by a diagonalization - It might be worth mentioning efficient ways to obtain also the eigenstates of each eigenvalue
Report
Requested changes
-
As far as I understand, a practical implementation immediately runs into the problem of knowing, a priori, how many states are contained in the filtered subspace. The authors repeatedly use the fact that their system contains ~5000 filtered eigenvalues, but this will not be known in general. This dimension should be estimated beforehand in order to avoid using too small a cutoff in the basis construction. Overestimating the cutoff however makes the second pass suboptimal. Hence, an accurate estimate seems important. How can that be obtained efficiently, without trial and error?
-
The relation between indices i, j and x,y in Eqs (23, 24) are not explicitly given, and confuses the reader. Either expand on the difference, or set i, j = x, y
-
The block filtering technique seems important, as near degeneracies are a common occurrence. A little clarification is requested. If one uses 5 independent random seeds for example, would then one get a 5x5 block from Eqs (23) and (24) for each x, y? Would one then simply build the final S and H from these 5x5 blocks? Can one still use x, y = i, j in this case?
-
Not critical for performance but: why is the final generalized eigenvalue problem not solved using the standard QZ factorization approach? Why not filter excess dimensions that way? Isn't it more efficient than a SVD followed by a diagonalization?
-
If one needs not only the eigenvalues but also the eigenstates, is there a smart way to get them without storing the full filtered basis in the second pass? This is an implementation optimization, response optional.
Author: Haoyu Guan on 2021-10-24 [id 1874]
(in reply to Report 1 by Pablo San-Jose on 2021-10-14)
We are grateful to the Referee for appreciating our work, for valuable advice and for constructive suggestions. We have revised the manuscript according to the requirements of both referees. We list the replies to the requested changes point by point.
1. Indeed, it is important to estimate the number of eigenvalues in the interval, and we have mentioned an estimation method in the second paragraph in Page 10. In the literature, there are several DOS estimation methods giving satisfactory results with high efficiency. For example, Ref. [43] in the manuscript (https://journals.aps.org/pre/abstract/10.1103/PhysRevE.62.4365); The kernel polynomial method is employed in EVSL (https://arxiv.org/abs/1802.05215) to estimate DOS before calculating eigenvalues. They are in general computationally inexpensive relative to computing the eigenvalues.
2. We have revised the manuscript to explicitly state the relationship between x and i. And the same equation holds for y and j.
3. We have revised the manuscript to illustrate the process of block filtering by a 2x2 matrix example. The relationship between x, y and i, j remains the same.
4. Since the final step is not critical in calculating the interior eigenvalues, we have just deduced it using our knowledges on linear algebra. We will try the QZ factorization method in the future. When the subspace dimension is tens of thousands, this may be critical. We thank the Referee for the advice.
5. For the moment we could only figure out a simple solution, that is to run the Chebyshev evolution once more. Since the unitary rotation matrix is not known before SVD, we don’t know how to combine the states in the original space to get the final eigenstates during the first run of Chebyshev evolution.
Anonymous on 2021-10-18 [id 1859]
How do you deal with the case where the filtered subspace is centered around zero energy (eigenvalue of the rescaled G in (23, 24))? In this particular case, it seems to me that when trying to build the subspace basis, the state from the first iteration (Psi_2 = G * Psi_E) is likely to be almost zero. Wouldn't this makes the generation of a subspace basis fail? As presented here, I think the recursive scheme would fail to efficiently generate a basis of the filtered subspace in this case.
Anonymous on 2021-10-18 [id 1860]
(in reply to Anonymous Comment on 2021-10-18 [id 1859])In this work we are trying to deal with exactly the energy centered around zero. And in fact, the Chebyshev polynomials on the interval [-1,1] are "the closest" to zero in the sense of polynomial space. The magical point is each one of T_n(x) behaves in a different mode, so different that they are orthogonal (ref. [49], Chapter 4) to each other. Moreover, since the basis states are normalized ones, the absolute maximum of the function is not involved, and the important thing is their oscillation modes.

---

## Round 2 · Referee Report · Anonymous (Referee 2) · 2021-10-20

Strengths
- interesting topic
- easy to understand
Weaknesses
- minor oddities in language
Report
Concerning the error analysis: I don't understand the concluding sentence of appendix E, "the main error source comes from the imperfect initial state $|\psi_E\rangle$ in Eq. (6), while the perfect one shall be ...".
Can this problem be fixed?
In addition, there are still a few oddities in language, e.g. page 11 center: "closely lied eigenvalues", or page 16 in the conclusion: "its computation time is almost irrelevant to the number of required eigenvalues and its memory requirements are irrelevant to the system sizes." Maybe some native speaker or editor can proofread the text.
Requested changes
- Clarify the last sentence of appendix E.
- Proofread once again.
Author: Haoyu Guan on 2021-10-24 [id 1875]
(in reply to Report 2 on 2021-10-20)We thank the Referee for the positive attitude towards our manuscript and for pointing out the weakness of the current version. In the new submitted version, we have revised the manuscript according to the suggestions of both referees. In particular, we have both clarified the last sentence of appendix E and proofread once again.

---

## Round 2 · List of Changes

1. We add discussions on other filtered interior eigensolvers in the third paragraph of Section 1, and stated in the next paragraph that instead of iterative filtering, the filtration in DACP is done only once.
2. We add Figure 2 (in the revised manuscript) to compare the exp-semicircle filter with two other polynomial filters. The second paragraph in Page 6 is also added to explain it.
3. We revise the second paragraph in Page 8 to further illustrate the motivation of employing the Chebyshev evolution.
4. We add Figure 4 to compare the level-spacing statistics of the results calculated with a single run of DACP and the Poissonian or Wigner-Dyson statistics. The associated discussion is added as the second paragraph in Page 11.
5. We add Section 4 in the revised version, to compare DACP with EVSL, FEAST and shift-invert methods in terms of both runtime and memory.
6. We add Figure 8 to test the basis function H^k instead of T_k(H) during the second step of DACP.
7. We add Appendix E to discuss the major error source and to estimate the mathematical expression of error distributions, with the visualized results shown in Figure 9.

---

## Round 3 · List of Changes

1. We revise the first paragraph of the Conclusion Section to obtain a clearer expression.
2. We add the relationship between x and i, and illustrate the process of block filtering by a 2x2 matrix example in Appendix C.
3. We revise the last sentence of Appendix E to make it clearer.
4. We have proofread the manuscript again.

---

## Editorial Decision

published